# The global phylogeography of rapidly expanding multidrug resistant Ural lineage 4.2 *Mycobacterium tuberculosis*

Melanie H. Chitwood [1] ✉, Isabel Rancu[1], Yexuan Song [2], Barney I. Potter[1], Yi Ting Chew [1], Nelly Ciobanu[3], Valeriu Crudu [3], Caroline Colijn [2], Ted Cohen [1,5] ✉ & Benjamin Sobkowiak [1,4,5]

Multidrug resistant tuberculosis (MDR-TB) epidemics are sustained by transmission of reproductively fit MDR *M. tuberculosis* (*Mtb*) strains. We search a large publicly available dataset of ~200,000 *Mtb* whole genome sequences to identify strains related to a highly successful MDR clade circulating in Moldova belonging to lineage 4.2.1/Ural. We characterize a clade of 1604 drug-resistant *Mtb* sequences harboring conserved resistance-conferring mutations. We identify the Russian Federation as the most likely country of origin for this clade and infer several independent migration events from Russia and Moldova to other European and Asian countries. We estimate that this clade is expanding more rapidly than comparable clades of lineage 4.2.1/Ural. The broad dispersal of this highly successful clade is an urgent global health threat. Genomic surveillance is essential to track the evolution and spread of this and other strains of concern.

Multidrug resistant tuberculosis (MDR-TB) is an emerging global health challenge for TB elimination efforts. While drug resistance-conferring mutations can arise over the course of treatment, transmission of drug-resistant *Mycobacterium tuberculosis* (*Mtb*) strains sustains MDR-TB epidemics in high burden settings[1]. Several recent studies have highlighted the role of MDR *Mtb* transmission in the Russian Federation and former Soviet Republics[2–5], where, in some settings, over 40% of new TB cases have a drug-resistant phenotype[6]. Of the ten human-adapted *Mtb* lineages, lineages 2 and 4 are the most strongly associated with MDR phenotypes, and the dominant multidrug resistant strains within local epidemics often have lineage 2 or 4 backgrounds[7]. Lineage 4 is genetically diverse and geographically widespread, supporting both globally represented "generalist" strains and geographically restricted "specialist" strains[8]. The lineage 4.2.1/Ural has been described as an "intermediate" strain; strains belonging to this sublineage have been identified in eastern European, central Asian[8,9], and east African countries[8], and may have the potential to spread widely.

Several recent studies have described a highly successful strain of MDR lineage 4.2.1/Ural *Mtb*. A study of lineage 4.2.1/Ural *Mtb* in eastern Europe identified an epidemic clone resistant to rifampin, isoniazid and kanamycin that had reached epidemic proportions in the Republic of Moldova within the last 25 years[10]. A study in the Moldovan capital of Chisinau described a similar MDR lineage 4.2.1/Ural strain that emerged in the 1990s and underwent significant expansion over the same time period[11]. Most recently, a country-wide *Mtb* phylogeographic analysis identified a large clade of the same Ural strain with evidence of high levels of transmission throughout Moldova[2,12]. This strain was estimated to have an effective reproduction number twice that of drug-susceptible lineage 4.2.1/Ural strains, and appeared to be expanding more rapidly than lineage 2 MDR strains in Moldova[13]. Global data suggest that the emergence of multidrug resistant lineage 4 strains is a local phenomenon, with limited evidence of migration of resistant strains across borders[14]. However, some Ural MDR-TB strains isolated in the Republic of Georgia[4] appear to be genetically similar to

[1]Department of Epidemiology of Microbial Diseases, Yale School of Public Health, New Haven, CT, USA. [2]Department of Mathematics, Simon Fraser University, Burnaby, BC, Canada. [3]Phthisiopneumology Institute, Chisinau, Republic of Moldova. [4]Department of Infection, Immunity and Inflammation, University College London, London, UK. [5]These authors contributed equally: Ted Cohen, Benjamin Sobkowiak. ✉e-mail: melanie.chitwood@yale.edu; theodore.cohen@yale.edu

the rapidly spreading Ural MDR-TB strains in Moldova[13], suggesting there may be more widespread dispersal of this lineage.

In this study, we constructed a global dataset of approximately 200,000 *Mtb* whole genome sequences available from public databases to assess the prevalence of lineage 4.2.1/Ural strains and identify strains genetically similar to the highly successful lineage circulating in the Republic of Moldova. Using genomic epidemiological analyses, we described the geographic spread, relative transmission fitness, and evolutionary history of this strain.

## Results

### Identification and global dispersion of lineage 4.2/Ural sequences

We analyzed 5909 *Mtb* whole genome sequences classified as lineage 4.2.1/Ural strains that were downloaded from the European Nucleotide Archive (ENA) (Supplementary Fig. 1). The country of origin was available for 5440 sequences (92%) and the date of specimen collection was available for 4062 sequences (69%). The completeness of date information varied: 1373 (23%) had complete dates, 2452 (41%) had partial dates, and 237 (4%) had a range of possible collection years. The oldest included sample was collected in 1994, and the newest included sample was collected in 2023. We identified sequences from 61 countries across 6 continents (Fig. 1). The countries with the largest share of lineage 4.2.1/Ural sequences were the Republic of Moldova (1546; 26%) and the Republic of Georgia (602; 10%). Notably, for several European countries, we did not identify any lineage 4.2.1/Ural sequences in publicly available datasets.

### Identification and dispersion of lineage 4.2.1.2

We inferred a maximum-likelihood phylogeny (Supplementary Fig. 2) and subsequently constructed a time-calibrated phylogeny by time-scaling branches using sampling dates. We identified a large clade (*n* = 1604) of MDR *Mtb* sequences that harbored clade-defining mutations (70 SNPs and 5 small insertions and deletions indels)[13] and were genetically similar to the previously identified MDR *Mtb* strain in Moldova[2,10,11,13] (Fig. 2). Strains with this genetic background have previously been called Ural Clade C[10], multidrug resistant outbreak strain[11], Ural Clade 1[2], and Ural_A[13]. The divergence of this clade from other Ural 4.2.1 strains may constitute its designation as a novel sublineage (e.g., lineage 4.2.1.2). Of the 70 clade-defining SNPs previously

identified[13], there were 28 synonymous SNPs found outside of drug resistance determining loci and PE/PPE gene families that may be incorporated into existing SNP barcoding schemes (Supplemental Table 1)[15]. We therefore refer to this clade as lineage 4.2.1.2.

All lineage 4.2.1.2 isolates had mutations associated with isoniazid (INH) resistance. Additionally, 1528 (95%) also carried mutations associated with rifampin (RIF) resistance and thus were MDR; this group included 399 pre-extensively drug resistant (XDR) isolates (MDR plus resistance to fluroquinolones [FLQ]) and 16 XDR isolates (pre-XDR plus resistance to one Group A drug, e.g., bedaquiline [BDQ] or linezolid [LZD]). The country of collection for sequences in lineage 4.2.1.2 was predominately Moldova (*n* = 1256; 78%), with sequences from eastern European and central and western Asian countries comprising 5% of the dataset (*n* = 78) and 10% from other European countries (*n* = 155). Based on the time of most recent common ancestor (tMRCA) of the timed clade, we estimated that lineage 4.2.1.2 emerged in 1971 (95% CI: 1965, 1976).

To characterize the movement of this strain across national borders, we inferred the country of origin for internal nodes of the lineage 4.2.1.2 clade using Sampling Aware Ancestral State Inference (SAASI)[16], an ancestral state inference method that explicitly accounts for sampling differences (Fig. 3). We infer that lineage 4.2.1.2 emerged first in Russia (root state probability = 0.98) and that Russia was the source country for 128 migration events (45%). Migration events from Russia to Moldova accounted for 73% of outflow from Russia and 72% of inflow to Moldova. We inferred a relatively small number of migration events into other eastern European and central and western Asian countries; of the 31 events, Russia was the source country for 17 (55%) events, and Moldova was the source country for 6 (19%). Moldova was the inferred country of origin in 96 total migration events; the majority (84 events, 88%) were migrations from Moldova into countries in western and southern Europe, primarily Germany (Supplementary Fig. 3A).

As a sensitivity analysis, we performed a conventional ancestral state reconstruction using the R package *ape* (which does not account for the sampling variability present in the data)[17]. Even when we ignored sampling variability, we found that Russia was the most likely source of lineage 4.2.1.2 (root state probability = 0.71). As expected, the method inferred a higher rate of migration events from Moldova (Supplementary Fig. 3B).

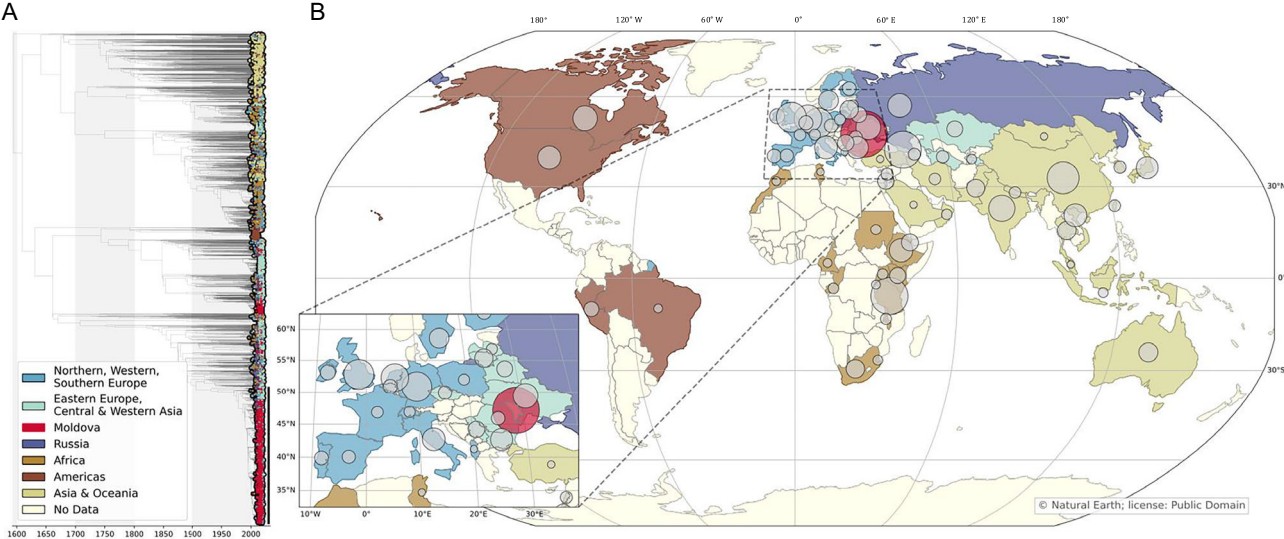

**Fig. 1 | Global Dispersion of lineage 4.2 Mtb strains. A** Time-calibrated phylogeny containing 5909 *Mtb* sequences included in the study; taxa are colored according to sequence region of origin. Lineage 4.2.1.2 is marked with a vertical black line. **B** Map of countries of origin for included sequences; circle is proportional to the number of sequences included from each country.

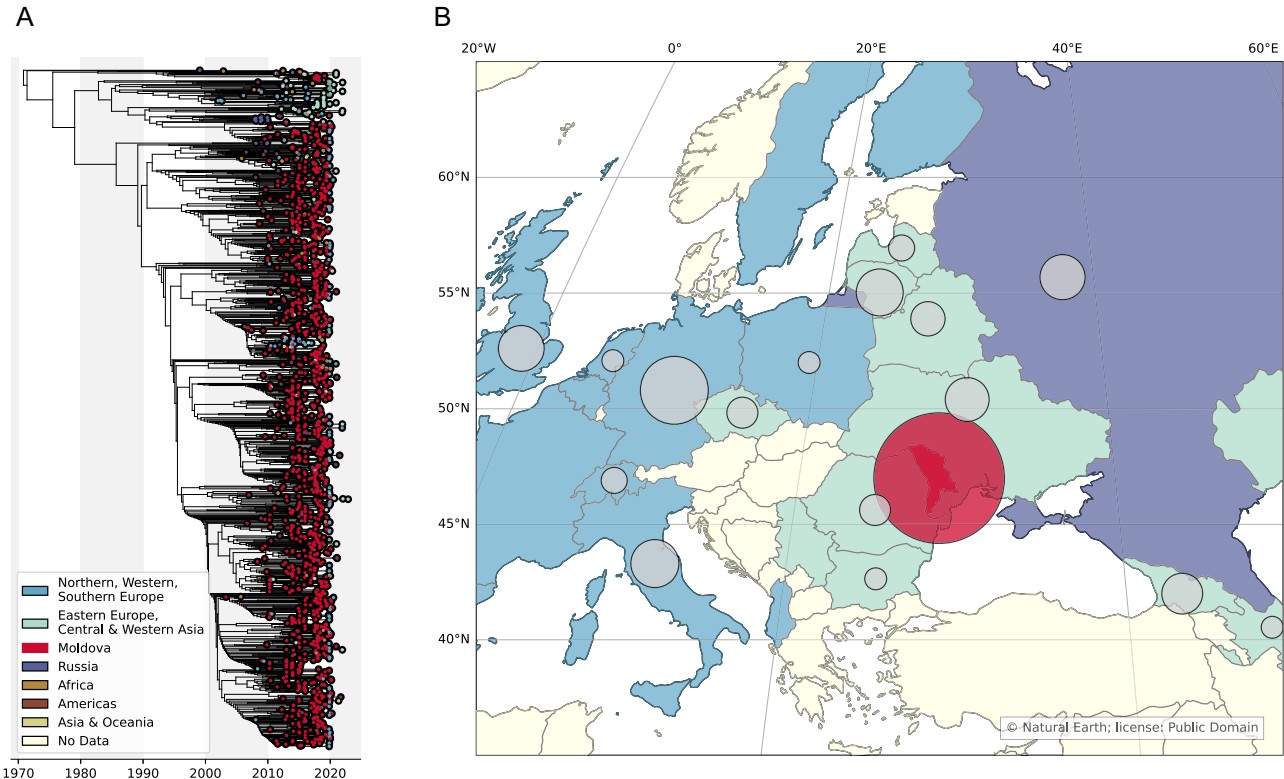

**Fig. 2 | Lineage 4.2.1.2 Phylogeny and Geographic Dispersion. A** Time-calibrated phylogeny containing 1604 *Mtb* sequences identified as part of lineage 4.2.1.2; taxa are colored according to sequence region of origin. **B** Map of countries of origin for sequences in lineage 4.2.1.2; circle is proportional to the number of sequences included from each country.

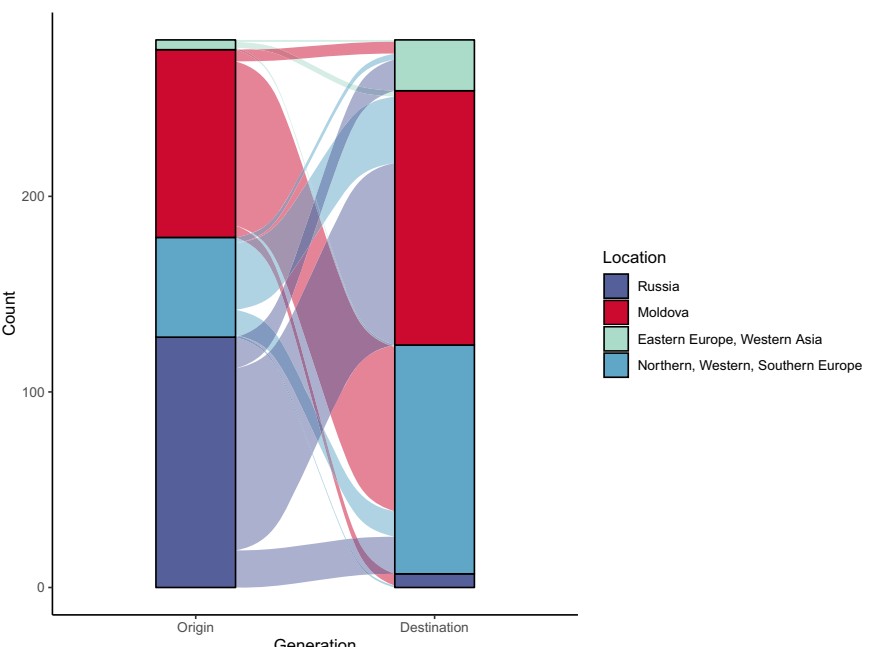

**Fig. 3 | Migration of Lineage 4.2.1.2 strains.** Alluvial plot showing the inferred origin and destination for each migration event. Sequences from countries with fewer than five isolates have been excluded. "Eastern Europe, Western Asia" includes Belarus, Georgia, and Ukraine; "Northern, Western, Southern Europe" includes Germany, Italy, Lithuania, Portugal, and the United Kingdom. Migration events between countries within these groups have been included in the plot.

## Evolution of drug resistance in lineage 4.2.1.2

We used ancestral state reconstruction to characterize the emergence and distribution of 24 key drug resistance-conferring mutations to eight antimicrobials (RIF, INH, ethambutol [ETH], FLQ, kanamycin [KAN], streptomycin [STR], LZD), as well as RIF compensatory mutations (Fig. 4). All lineage 4.2.1.2 isolates carried the *katG* Ser315Thr mutation that confers resistance to INH. Of those, 1442 (90%) also had an *inhA*.−777C > T mutation (also referred to as *fabG1*.−15C > T) in the

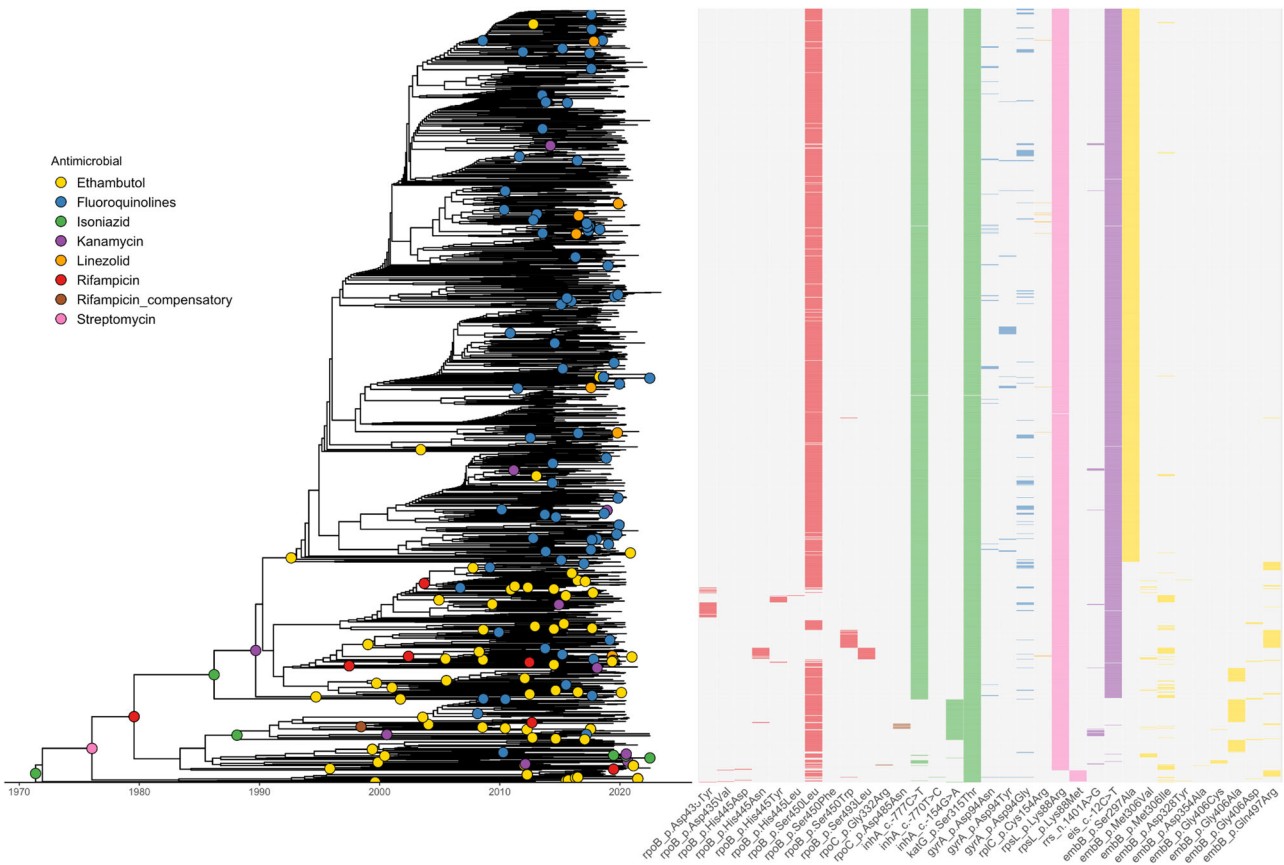

**Fig. 4 | Ancestral state reconstruction of drug resistance-conferring mutations.** Phylogenetic tree of lineage 4.2.1.2 with the inferred drug resistance genotypes.

upstream regulatory region of the *fabG1-inhA* operon, which confers low-level resistance to INH and was predicted to have emerged in the mid 1980s. A smaller number of other isolates (84; 5.2%) instead carried the *inhA*.−154G > A mutation in the same regulatory region, which emerged in the late 1980s.

Most isolates (1406; 88%) harbored the *rpoB* Ser450Leu mutation conferring RIF resistance, which emerged in lineage 4.2.1.2 in the late 1970s. Of the remaining isolates, 123 contained a single other RIF resistance-associated mutation in *rpoB*, and 21 strains had a double mutation in *rpoB* His445Asn and Ser493Leu that conferred RIF resistance. These mutations emerged several times from the late 1990s onwards. There were also 12 isolates that had mutations in *rpoC* that have previously been associated with compensatory mechanisms to RIF resistance (*rpoC* Gly332Arg[18] and Asp485Asn[19]) and emerged in the late 1990s in the clade.

We infer that resistance to STR first evolved in the clade in the mid 1970s as a result of the *rpsL* Lys88Arg mutation, which was carried by 1579 isolates (98%); 23 of the 25 remaining isolates later acquired the alternative *rpsL* Lys88Met mutation at the same locus. From the mid 2000s onwards, there were many occurrences of three FLQ resistance-conferring mutations evolving independently in the *gyrA* gene. Additionally, 1427 (89%) isolates carried the *eis*.−12C > T mutation−associated with resistance to KAN−that emerged around 1990, with 14 of these also possessing the *rrs*. 1401 A > G KAN resistance-conferring mutation and 14 additional KAN resistant isolates carrying only this *rrs* mutation. Finally, we found that resistance to newer antimicrobials was uncommon in lineage 4.2.1.2. Recent evolution of LZD resistance was identified in 11 isolates that carried the *rplC* Cys154Arg mutation, and four isolates contained a duplication in *mmpR5* that is associated with BDQ resistance[20].

## Recent expansion

We compared the distribution of the local branching index (LBI)[21] of taxa in lineage 4.2.1.2 to taxa in comparison clades. LBI is a measure of relative transmission fitness based on the topology of the phylogenetic tree. High fitness ancestors (internal nodes) will produce more rapid branching patterns in the phylogeny, and sampled isolates (taxa) of higher reproductive fitness can be identified as their recent descendants.

Four other lineage 4.2.1/Ural clades with at least 150 taxa and a tMRCA within 50 years of the emergence of lineage 4.2.1.2 were used for comparisons: clade 1 (tMRCA = 1960 [1954, 1967], *n* = 212), clade 2 (tMRCA = 1949 [1945, 1953], *n* = 251), clade 3 (tMRCA = 1987 [1982, 1991], *n* = 185), and clade 4 (tMRCA = 1945 [1937, 1952], *n* = 152) (Supplementary Fig. 4). We found that lineage 4.2.1.2 had a higher median LBI than the taxa in other clades (Tukey test *p* value < 0.001 for all comparisons to other clades) (Fig. 5a). We also found that, across countries, lineage 4.2.1.2 had a higher LBI on average than taxa in comparison clades (Fig. 5b).

Within lineage 4.2.1.2, the LBI for taxa from Russia did not differ significantly from those of other eastern European countries (excluding Moldova) and central and western Asian countries (Tukey test *p* value = 0.48). However, taxa from Moldova had a higher average LBI than taxa from other eastern European (excluding Russia) and central and western Asian countries (Tukey test *p* value < 0.001) and from Russia (Tukey test *p* value < 0.001). In eastern European and central and western Asian countries (excluding Russia and Moldova), we observed a bimodal distribution of LBI within lineage 4.2.1.2. Most sequences in this group come from Georgia and Ukraine; strains from Georgia had a lower LBI on average than those from Ukraine (Tukey test *p* value < 0.001).

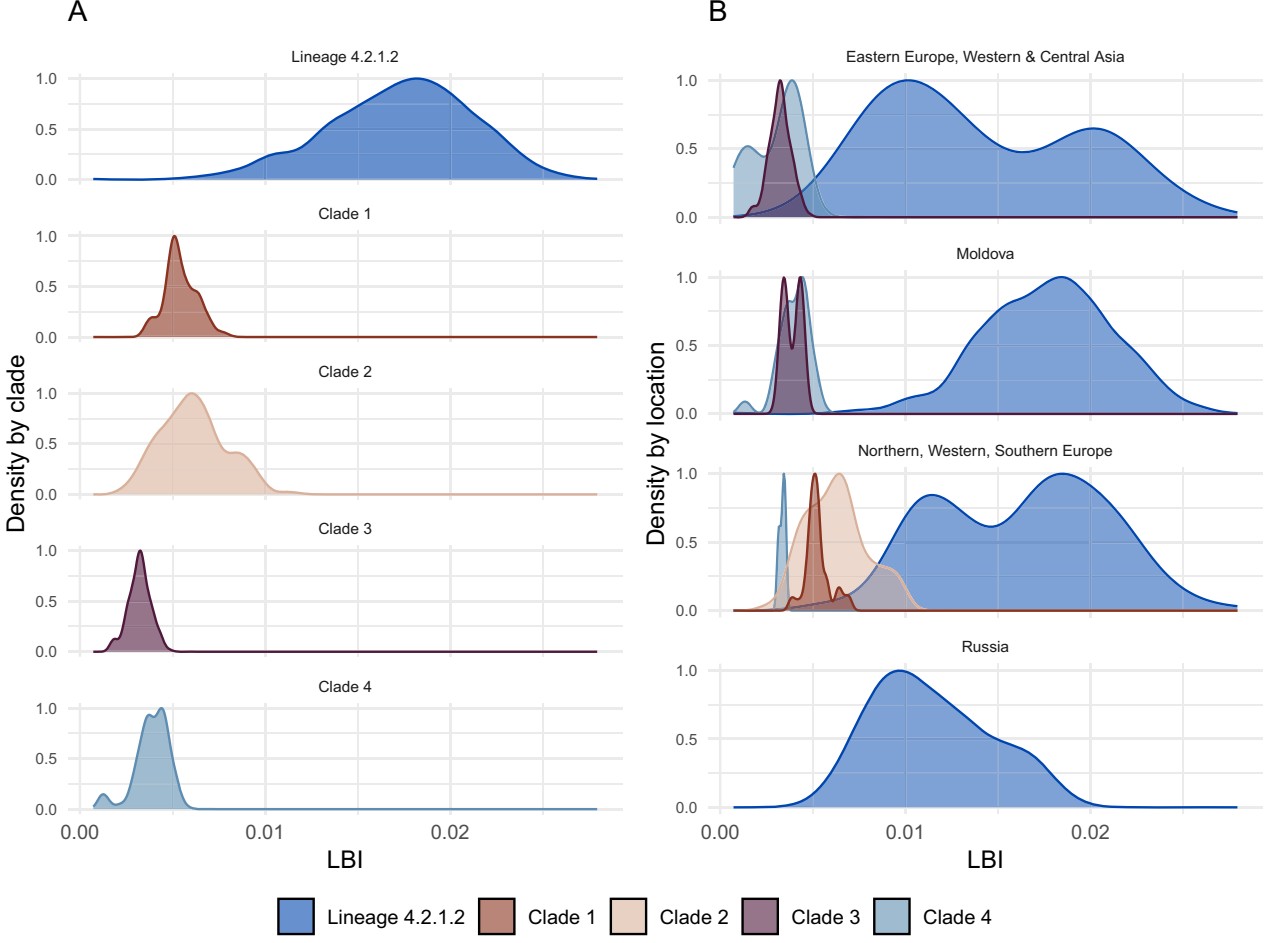

**Fig. 5 | Local Branching Index. A** Distribution of LBI for taxa in lineage 4.2.1.2 and comparison clades. **B** Distribution of LBI for taxa in lineage 4.2.1.2 and comparison clades by country or region of origin, colored by clade.

## Discussion

We identified almost 6000 *Mtb* sequences belonging to the *Mycobacterium tuberculosis* complex lineage 4.2.1/Ural in publicly available sequence repositories. Among those sequences, 1604 belonged to a large clade of drug-resistant *Mtb* that contains MDR *Mtb* strains from Moldova with a high effective reproduction number[13]. Using LBI, we estimated that lineage 4.2.1.2 is growing more rapidly than other *Mtb* lineage 4.2.1/Ural clades emerging over similar time periods.

Our analysis suggested that lineage 4.2.1.2 likely emerged in Russia around 1971 and subsequently spread throughout Europe and central and western Asia. This finding differs from an earlier analysis that suggested similar strains emerged in Moldova and spread to neighboring countries, including Georgia and Russia[11]. In that study, the authors included a limited number of sequences from outside of Moldova and identified Moldova as the country of origin with only moderate certainty (posterior probability = 0.66). Our analysis included a larger sample of isolates from a broader set of countries, and we identified Russia as the country of origin with a high degree of certainty (probability = 0.98). Both studies conclude that there has been subsequent spread of this strain from Moldova into neighboring countries. However, our analysis suggests that migration events out of Russia also played an important role in the international spread of lineage 4.2.1.2.

We found evidence that most MDR isolates in lineage 4.2.1.2 carried the same *rpoB* mutations and all had a fixed *katG* mutation; these confer resistance to rifampin and isoniazid, respectively. Ancestral state reconstruction suggested that these mutations first evolved in the 1970s and were conserved in subsequent generations. Conversely,

we found that the evolution of fluroquinolone resistance was likely driven by recent, independent acquisitions of mutations in the *gyrA* gene. Rifampin and isoniazid are first-line drugs that have been used to treat TB for decades; mutations that confer resistance to these first-line drugs have had many years of selective pressure pushing them towards fixation. Fluroquinolones are newer, second-line drugs used to treat individuals with MDR-TB, and there have been fewer opportunities for positive selection to favor mutations which confer resistance to these antimicrobials.

We attempted to include every publicly available lineage 4.2,1 *Mtb* sequence from ENA. This allowed us to describe the global distribution of lineage 4.2.1 strains and identify migration events in greater detail than in previous analyses[14]. In some settings, isolates were collected as part of a city- or country-wide prospective whole genome sequencing study[19,22], though in other settings sequences were part of dedicated studies on MDR-TB[23,24]. Because of the variable sampling strategies, it is challenging to fit phylodynamic models to these data. In the case of the country-level ancestral state reconstruction, we were able to overcome this challenge by using an approach that adjusts for heterogenous sampling (SAASI, see Methods). However, this still presents a limitation when determining the extent to which MDR lineage 4.2.1 isolates are present outside of Moldova, where there have been several large sequencing studies[2,10,11].

We were also limited by the availability of strain metadata; for example, data was not available on the country of origin for 8% of isolates or the date of isolation for 31% of isolates. Finally, several European countries had no available lineage 4.2.1 *Mtb* sequences. In

many of these settings, we are aware of the existence of *Mtb* whole genome sequence data that have not been made publicly available (e.g. EuSeqMyTB).

We present evidence to suggest that a rapidly expanding strain of MDR *Mtb*, which was previously believed to be restricted to the Republic of Moldova, has spread to other European and central and western Asian countries. The broad dispersal of lineage 4.2.1.2 is an urgent threat to TB control in the region. While rapid molecular tests can quickly identify drug-resistant disease, they cannot identify specific bacterial lineages or strains. Routine whole genome sequencing is therefore essential to support surveillance of lineage 4.2.1.2 and other strains of concern.

## Methods

### Global collection of Mtb Ural lineage 4.2.1 whole genome sequence data

We queried ENA on 18 February 2024 for all *M. tuberculosis* genomes ($n = 196,547$ accessions). ENA is synchronized with GenBank, making it the most complete public source of whole genome sequencing data for *Mycobacterium tuberculosis* complex (MTBC). We excluded laboratory and reference strains, other Mycobacteria species in the MTBC, and samples isolated from non-human hosts. We compared the remaining sample accession numbers ($n = 177,856$) in ENA to the TB-Profiler (TBP) database[25], which contains lineage assignments and drug resistance predictions for publicly available *Mtb* sequences on the Sequence Read Archive (https://www.ncbi.nlm.nih.gov/sra). While the TBP dataset is updated regularly, we found a subset of sample accession numbers from ENA that did not already have an established lineage or sub-lineage assignment in this database ($n = 41,233$). These sequences were subsequently profiled by downloading the sequencing files from ENA and running TB-Profiler. We identified 7165 unique sample accessions that were profiled as *Mtb* lineage 4.2.1, which comprised 7563 whole genome sequencing data files (including samples with duplicate sequencing data or that were re-sequenced).

The 7563 sequencing data files were downloaded from ENA and aligned to the H37Rv reference strain (NC_000962.3) using BWA-MEM[26] for both paired and single end read data. Binary alignment (BAM) files were indexed and sorted with SAMtools[27]. Alignments with less than 80% mapping to the H37Rv reference strain and an average read depth below 50x were removed, along with any sample with evidence of mixed infection detected using MixInfect2[28]. In cases where samples had multiple run accessions (duplicate or re-sequenced isolates), alignments with the highest mapping and average read depth were retained for a final dataset of $n = 5909$ *Mtb* lineage 4.2.1 sequences (one clinical sequence per sample specimen) (Supplementary Data 1).

Variant calling was conducted using GATK[29] "HaplotypeCaller" and "GenetypeGVCFs"; low-confidence variants ($Q < 20$, read depth < 5) and sites with an ambiguous or missing call in more than 10% of isolates were removed. The consensus nucleotide ($\geq 80\%$ of mapped reads) was assigned at loci with mixed calls, otherwise the nucleotide 'N' was assigned. Finally, variants in repetitive regions, in PE/PPE genes, and at known resistance-conferring loci, were removed. A multi-sequence alignment of variant SNPs was constructed for subsequent analyses.

We cross-referenced sample country and collection date between ENA and TBP. In cases of conflicting country metadata, we preferentially used the value from ENA. For conflicting dates, we used the most complete date available; when both dates were complete, we used the date recorded in ENA. For isolates belonging to large projects (10 or more sequences included in this study) with missing collection country or date, we queried PubMed for publications associated with the BioProject ID. If metadata were not available as a supplement to these studies, we requested these data from corresponding authors.

### Phylogenetic reconstruction

We performed maximum-likelihood phylogenetic reconstruction using *IQ-TREE 2*[30] from a multi-sequence alignment of concatenated SNPs. The optimal substitution model (TVM + F + G4) was determined using the model test ("-m") option, and branch support was calculated using 1000 bootstrap replicates. We then performed Bayesian inference of a time-calibrated phylogenetic tree using the R package *BactDating*[31], fitting the model using the maximum-likelihood phylogeny after scaling the branch lengths to SNPs/genome/year, and calibrating the tree using sampling dates. Where sampling dates were uncertain or unavailable, we used uniform priors with bounds indicating the earliest and latest possible sampling date. We fit the model using a fixed mean clock rate of 0.5 SNPs/genome/year (approx. $1.145 \times 10^{-7}$ SNPs/site/year)[32] and a strict gamma clock model. We ran the model for $5 \times 10^5$ MCMC samples, thinning the posterior by a factor of 500, resulting in 1000 posterior samples.

We characterized a monophyletic clade within the time-calibrated phylogeny that included all isolates previously identified as part of a rapidly expanding MDR-TB strain in Moldova that harbored clade-defining mutations; 70 SNPs and 5 small insertions and deletions (indels)[13]. Finally, we used the R package *treestructure*[33] to identify comparable lineage 4.2.1 clades, each with 150 or more taxa and a tMRCA within 50 years of lineage 4.2.1.2's most recent common ancestor.

### Phylodynamic and genomic analyses

We performed ancestral state reconstruction using SAASI, an ancestral state inference method that explicitly accounts for sampling differences and is computationally feasible on large trees[16]. We used SAASI to infer the ancestral states of lineage 4.2.1.2 strains from Belarus ($n = 6$), Georgia ($n = 12$), Germany ($n = 92$), Italy ($n = 23$), Lithuania ($n = 21$), Moldova ($n = 1256$), Portugal ($n = 8$), Russia ($n = 18$), Ukraine ($n = 16$), and the United Kingdom ($n = 19$).

SAASI requires estimates of the branching rate and the removal rate (termed the "speciation" and "extinction" rates, with reference to the evolution literature in which state-dependent speciation and extinction models were conceived[34]), as well as the rates of transition among states (geographic regions) and sampling rates. We estimated the speciation (estimate: 0.174) and extinction (estimate: 0.001) rates using a maximum-likelihood approach[35], assuming that the sampling rate is known. We estimated the transition rates between different countries using *ace* in the *ape* package in R[17]. We specified a three-parameter model: (i) a transition rate from Moldova to other countries (estimate: 0.001), (ii) a transition rate from other countries to Moldova (estimate: 0.009), (iii) a transition rate from any pair of the non-Moldova countries (estimate: 0.004).

Finally, we estimated sampling rates by first estimating sequencing coverage by country and then scaling that to the average sequencing coverage in the clade and the inferred speciation rate. We use the following equations:

$$\psi_c = \frac{S_c}{F_c \cdot W_c} \qquad (1)$$

$$a = \frac{\sum (\psi_c \cdot S_{L4.2.1.2,\, c})}{\sum S_{L4.2.1.2,\, c}} \qquad (2)$$

$$\frac{\psi_c}{a} \cdot M \cdot b \qquad (3)$$

where $S_c$ is the observed number of lineage 4.2.1 sequences, $W_c$ is the estimated number of TB cases using WHO notification data, $F_c$ is the estimate fraction of cases belonging to lineage 4.2.1. We estimate this fraction using data from 2015–2019, the same period in which 50% of

the sequences in the clade were collected. We assumed 10% sequencing coverage when data on TB incidence were unavailable, and we assumed no country had a sequencing coverage >50%. In a sensitivity analysis we assumed 5% sequencing coverage when data on TB incidence were unavailable; we found that our results were not overly sensitive to this choice (Supplementary Fig. 5). We normalize $\psi_c$ by $a$, the weighted average sequencing coverage in the lineage 4.2.1.2 clade. Since this results in a fraction, not a rate per unit time, we multiply this value by the inferred speciation rate $M$, scaled by a factor $b$ (chosen such that no sampling rate exceeds the inferred speciation rate; $b = 0.8$).

The emergence of key drug resistance mutations in lineage 4.2.1.2 was inferred using a maximum-likelihood marginal reconstruction of ancestral sequences at nodes in the timed phylogeny, implemented in the R package *Phangorn*[36]. We included ambiguous sites and missing calls to reflect prior probabilities of all character states. Mutations conferring resistance to RIF, INH, ETH, FLQ, KAN, STR, and LZD, along with rifampin resistance compensatory mutations, were determined using the WHO catalogue[37].

Finally, we calculated LBI[21] at every node in the maximum likelihood phylogeny of lineage 4.2.1 ($n = 11,817$, taxa = 5909) using a neighborhood size of $2.18 \times 10^{-4}$ (0.0625 times the average pairwise patristic distance [$3.5 \times 10^{-3}$ substitutions/site]). We report the distribution of LBI for each terminal node. We compare the distribution of LBI across groups using Tukey's test for multiple comparisons.

This manuscript follows the STROME-ID guidelines[38].

## Reporting summary

Further information on research design is available in the Nature Portfolio Reporting Summary linked to this article.

## Data availability

All data were accessed from the European Nucleotide Archive. Run accession numbers, country of origin, and inferred sample dates are available in Supplementary Data 1.

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

## Acknowledgements

The authors report funding from the National Institutes of Health (R01AI180209: M.H.C., B.P., T.C., B.S., and P01AI159402: M.H.C., T.C., and B.S.), and the Medical Research Council (UKRI1414: B.S.).

## Author contributions

C.C. and B.S. conceived the study. IR and YTC assembled the data. M.H.C., Y.S., and B.S. analyzed the data. M.H.C., Y.S., B.I.P., and B.S. visualized results. T.C. secured funding. M.H.C., I.R., Y.S., B.I.P., Y.T.C., N.C., V.C., C.C., T.C., and B.S. reviewed results, contributed to manuscript drafting, and revised the manuscript.

## Competing interests

The authors declare no competing interests
