## [Peer Review file · Nature Communications]

The global phylogeography of rapidly expanding multidrug resistant Ural lineage 4.2 *Mycobacterium tuberculosis*

Corresponding Author: Dr Melanie Chitwood

Version 0:

Reviewer comments:

Reviewer #1

(Remarks to the Author)
COMMENTS TO AUTHOR

In general, the reported findings are partly novel. In brief, L4.2 originated in northern Eurasia (present Russia) and its MDR clone originated in Moldova. This is known from previous studies.

L4.2 sublineage (Ural family) of *M. tuberculosis* is known to be endemic in Russia (doi: 10.1016/j.meegid.2011.09.026 – that was the first study on the Ural family, although partly outdated; followed by: doi: 10.1016/j.tube.2015.02.031). Its MDR sublineage is actively circulating in Moldova and was first described as such by Sinkov et al. 2018 (that was the first paper that brought attention to this MDR clade in Moldova).

The particular interest of this study that the authors use very large dataset to decipher the origin of the “Moldovan” strain to be in Russia.

The text in Abstract is confusing: the authors confuse L4.2 lineage on the whole and its MDR clade/clone/strain (emerging in Moldova). It would be interesting and useful if you propose a barcode for this MDR clade within L4.2. Perhaps you may correlate with known barcode systems (Napier et al.; Shitikov et al.).

Reviewer #2

(Remarks to the Author)

This article provides an interesting overview of global phylogeography and spread of MDR Ural/Lineage 4.2 *Mtb* isolates. It reads well and provides confident evidence of the dispersal of Ural lineage. However, it doesn't mention some preliminary studies that have been done regarding Ural lineage (such as this study: <https://doi.org/10.1016/j.meegid.2011.09.026>). Furthermore, please explain better the approach used in line 224 “an approach that adjusts for heterogeneous sampling”.

Reviewer #3

(Remarks to the Author)

The paper “The global phylogeography of rapidly expanding multidrug resistant Ural lineage 4.2 *Mycobacterium tuberculosis*” by Melanie H Chitwood et al is an important and interesting paper on an MTB MDR strain which has spread efficiently in Moldova and beyond in recent decades. The paper is elegant and convincing and represents a substantial addition to the field. I still have some comments, mainly related to clarity of writing and methods used.

- General & intro

I think the geographic categories need to be cleaned up a bit. I would suggest using UN geoschemes as a guiding star, which accurately describes European countries as Western, Northern, Southern and Eastern. Here categories are thrown around a bit, and I agree using categories clearly is challenging, especially when additional political dynamics such as current EU, former Soviet etc are added to the mix.

One example: In Fig 2 Poland is categorized as “other European” and not as “former Soviet”. I guess this confusion stems

from the fact that Poland is today an EU member, and as such a fully integrated European country. It was however formerly occupied by the Soviet and is hence ALSO a former Soviet country/colony.

Lines 46-48. I think Eldholm et al PNAS 2016, which demonstrates that AMR evolution specifically occurred in former Soviet republics and not e.g. in Central Asian clades of the same lineage would be good to cite here.

- Methods:

Line 251 onwards. I would like some more details on how the TBP database was used here. How exactly were 177 856 genomes compared to the TBP database? Is this an open access resource? I know TBP but its pretty impressive to screen 177 856 genomes (!) Mustnt the genomes be downloaded pre screening?

Line 290. The authors fix the rate at 0.5 subst/genome/year (1.145×10^{-7} per site/year). I think this rate is sensible, but it should be backed up by citing relevant L4 phylodynamic papers. Other rate estimates in L4 are e.g. 4.84×10^{-8} (Brynildsrud et al Sci Adv 2018), 0.29 subst/genome/year (Eldholm et al Nat Comms 2014), 0.4 subst/genome/year (Roetzer et al Plos Med 2013), and 0.5 subst/genome/year (Walker et al Lancet Infect Dis 2013). Regardless, I think its important to keep in mind that fixing the rate like this will make TMRCA estimates unreliable, but as exact timing of TMRCA is not central to the paper, I think fixing the rate is acceptable, if backed up by the literature.

Lines 313 onwards. Please explain what speciation and extinction means in this context?

Line 327. A sampling rate /seq coverage of 10% seems very high. Are the inferences made sensitive to small adjustments of this parameter?

Version 1:

Reviewer comments:

Reviewer #3

(Remarks to the Author)

I am happy with the manuscript in its current state, and would like to commend the authors for a fine piece of work.

REVIEWER COMMENTS

Reviewer #1 (Remarks to the Author):

In general, the reported findings are partly novel. In brief, L4.2 originated in northern Eurasia (present Russia) and its MDR clone originated in Moldova. This is known from previous studies.

L4.2 sublineage (Ural family) of *M. tuberculosis* is known to be endemic in Russia (doi: 10.1016/j.meegid.2011.09.026 – that was the first study on the Ural family, although partly outdated; followed by: doi: 10.1016/j.tube.2015.02.031). Its MDR sublineage is actively circulating in Moldova and was first described as such by Sinkov et al. 2018 (that was the first paper that brought attention to this MDR clade in Moldova).

We have now included the Mokrousov citation in the Introduction. We note that we have already cited Sinkov et al. 2018 as the first study to report on this MDR strain circulating in Moldova.

The particular interest of this study that the authors use very large dataset to decipher the origin of the “Moldovan” strain to be in Russia.

The text In Abstract is confusing: the authors confuse L4.2 lineage on the whole and its MDR clade/clone/strain (emerging in Moldova). It would be interesting and useful if you propose a barcode for this MDR clade within L4.2. Perhaps you may correlate with known barcode systems (Napier et al.; Shitikov et al.).

Thank you for this suggestion. We have clarified the abstract to better differentiate between our MDR-TB clade of interest and lineage 4.2.1 broadly. We have also renamed our clade according to the barcode system you have referenced; we now refer to it as lineage 4.2.1.2 and include lineage-defining SNPs (first reported in Chitwood et al. Nat. Comms 2024) in Supplemental Table 1.

“The divergence of this important clade from other Ural 4.2.1 strains may constitute its designation as a novel sub-lineage (e.g., lineage 4.2.1.2). Of the 70 clade-defining SNPs previously identified,⁽¹³⁾ there were 28 synonymous SNPs found outside of drug resistance determining loci and PE/PPE gene families that may be incorporated into existing SNP barcoding schemes (Supplemental Table 1).⁽¹⁵⁾ We therefore refer to this clade as lineage 4.2.1.2.”

Reviewer #2 (Remarks to the Author):

This article provide interesting overview of global phylogeography and spread of MDR Ural/Lineage 4.2 Mtb isolates. It reads well and provide confident evidences of the dispersal of Ural lineage. However, it doesn't mention some preliminary studies that has been done regarding Ural lineage (such as this study: <https://doi.org/10.1016/j.meegid.2011.09.026>).

Thank you for this suggestion. We have added this citation to the Introduction.

Furthermore, please explain better the approach used in line 224 "an approach that adjusts for heterogenous sampling".

We have clarified that this sentence refers to SAASI, which is described in detail in the Methods section.

Reviewer #3 (Remarks to the Author):

The paper "The global phylogeography of rapidly expanding multidrug resistant Ural lineage 4.2 Mycobacterium tuberculosis" by Melanie H Chitwood et al is an important and interesting paper on an MTB MDR strain which has spread efficiently in Moldova and beyond in recent decades. The paper is elegant and convincing and represents a substantial addition to the field. I still have some comments, mainly related to clarity of writing and methods used.

- General & intro

I think the geographic categories need to be cleaned up a bit. I would suggest using UN geoschemes as a guiding star, which accurately describes European countries as Western, Northern, Southern and Eastern. Here categories are thrown around a bit, and I agree using categories cleanly is challenging, especially when additional political dynamics such as current EU, former Soviet etc are added to the mix.

One example: In Fig 2 Poland is categorized as "other European" and not as "former Soviet". I guess this confusion stems from the fact that Poland is today an EU member, and as such a fully integrated European country. It was however formerly occupied by the Soviet and is hence ALSO a former Soviet country/colony.

Thank you for this suggestion. We have updated our country groups and naming scheme to align the UN geoschemes. We've chosen to group Eastern Europe with Central and Western Asia (this keeps many of the Former Soviet Republics together) and to group Northern, Western, and Southern Europe together; this largely maintains the group previously called 'other Europe'.

Lines 46-48. I think Eldholm et al PNAS 2016, which demonstrates that AMR evolution specifically occurred in former Soviet republics and not e.g. in Central Asian clades of the same lineage would be good to cite here.

Thank you for this suggestion. We have added this citation to the Introduction.

- Methods:

Line 251 onwards. I would like some more details on how the TBP database was used here. How exactly were 177 856 genomes compared to the TBP database? Is this an open access resource? I know TBP but its pretty impressive to screen 177 856 genomes (!) Mustnt the genomes be

downloaded pre screening?

TB-Profiler periodically profiles the lineage and drug resistance of publicly available *Mtb* sequences from the Sequence Read Archive (SRA) to benchmark barcoding performance. A resulting dataset containing the sample accession, MTBC lineage and sub-lineage and drug-resistance profile is available on the TB- Profiler website (<https://tbd.r.lsh.t.m.ac.uk/sra>). However, not all accessions labelled as “Mycobacterium tuberculosis” in ENA were present in the TB- Profiler dataset (n = 41,233). Therefore, these sequences were downloaded from ENA and profiled using TB-Profiler, which can take the raw sequencing data files (FASTQ) as input. We have clarified the methods as follows:

“We compared the remaining sample accession numbers (n = 177,856) in ENA to the TB-Profiler (TBP) database, which contains a dataset of lineage assignments and drug resistance predictions for publicly available *Mtb* sequences on the Sequence Read Archive (<https://www.ncbi.nlm.nih.gov/sra>). While the TBP dataset is updated regularly, we found a subset of sample accession numbers from ENA that did not already have an established lineage or sub-lineage assignment in this database (n = 41,233). These sequences were subsequently profiled by downloading the sequencing files from ENA and running TB-Profiler. We identified 7165 unique sample accessions that were profiled as *Mtb* lineage 4.2, which comprised 7563 whole genome sequencing data files (including samples with duplicate sequencing data or that were re-sequenced).”

Line 290. The authors fix the rate at 0.5 subst/genome/year (1.145×10^{-7} per site/year). I think this rate is sensible, but it should be backed up by citing relevant L4 phylodynamic papers. Other rate estimates in L4 are e.g. 4.84×10^{-8} (Brynildsrud et al Sci Adv 2018), 0.29 subst/genome/year (Eldholm et al Nat Comms 2014), 0.4 subst/genome/year (Roetzer et al Plos Med 2013), and 0.5 subst/genome/year (Walker et al Lancet Infect Dis 2013). Regardless, I think its important to keep in mind that fixing the rate like this will make TMRCA estimates unreliable, but as exact timing of TMRCAs is not central to the paper, I think fixing the rate is acceptable, if backed up by the litterature.

We have included the Walker citation to support our use of 0.5 SNPs/genome/year. In addition, we tested fitting the model with slower clock rates; the estimated MRCA years were 1937 when we used 0.29 SNPs/genome/year (Eldholm et al) and 1957 when we used 0.4 SNPs/genome/year (Brynildsrud et al).

Lines 313 onwards. Please explain what speciation and extinction means in this context?

We have added new text to clarify these terms:

“SAASI requires estimates of the branching rate and the removal rate (termed the "speciation" and "extinction" rates, with reference to the evolution literature in which state-dependent speciation and extinction models were conceived), as well as the rates of transition among states (geographic regions) and sampling rates.”

Line 327. A sampling rate /seq coverage of 10% seems very high. Are the inferences made sensitive to small adjustments of this parameter?

We refit SAASI assuming a sequencing coverage of 5% when data on TB incidence were unavailable. We found that 5% sequencing coverage did not meaningfully change our results. We have now included text about this sensitivity analysis in the Methods section and the result are presented in Supplementary Figure 5:

Alluvial plot showing the inferred origin and destination for each migration event. Sequences from countries with fewer than five isolates have been excluded. “Eastern Europe, Western Asia” includes Belarus, Georgia, and Ukraine; “Norther, Western, Southern Europe” includes Germany, Italy, Lithuania, Portugal, and the United Kingdom. Migration events between countries within these groups have been included in the plot.